# GOAL-CONDITIONED VIDEO PREDICTION

## ABSTRACT

Many processes can be concisely represented as a sequence of events leading from a starting state to an end state. Given raw ingredients, and a finished cake, an experienced chef can surmise the recipe. Building upon this intuition, we propose a new class of visual generative models: goal-conditioned predictors (GCP). Prior work on video generation largely focuses on prediction models that only observe frames from the beginning of the video. GCP instead treats videos as start-goal transformations, making video generation easier by conditioning on the more informative context provided by the first and final frames. Not only do existing forward prediction approaches synthesize better and longer videos when modified to become goal-conditioned, but GCP models can also utilize structures that are not linear in time, to accomplish hierarchical prediction. To this end, we study both auto-regressive GCP models and novel tree-structured GCP models that generate frames recursively, splitting the video iteratively into finer and finer segments delineated by subgoals. In experiments across simulated and real datasets, our GCP methods generate high-quality sequences over long horizons. Tree-structured GCPs are also substantially easier to parallelize than auto-regressive GCPs, making training and inference very efficient, and allowing the model to train on sequences that are thousands of frames in length. Finally, we demonstrate the utility of GCP approaches for imitation learning in the setting without access to expert actions. Videos are on the supplementary website: `https://sites.google.com/view/video-gcp`

## 1 INTRODUCTION

Many phenomena, both natural and artificial, are naturally characterized as transformations — the most salient information about them is contained in the start and end states, given which it is possible to fill in intermediate states from prior experience. For example, ending up in San Francisco after starting in Oakland entails getting into a car and crossing the Bay Bridge. Similarly, to an expert engineer observing a bridge, the task of reverse-engineering how it was built is well-defined and tractable.

In contrast, consider the task of predicting forward in time, having observed only the steel and concrete that went into making the bridge. Such forward prediction tasks are severely underconstrained, leading to high uncertainties that compound with time, making it impossible to make meaningful predictions after only a few stages of iterative forward prediction (see Fig. 1). This is aggravated in high-dimensional settings such as forward *video* prediction, which despite being the most widely studied setting for video synthesis, struggles to produce coherent video longer than a few seconds.

We propose to condition video synthesis instead on the substantially more informative context of the start *and* the goal frame. We term such models goal-conditioned predictors (GCP). Much like the engineer observing the bridge, GCPs treat long videos as start-goal transformations and reverse-engineer the full video, conditioned on the first and final frames. The simplest instantiation of GCPs modifies existing forward prediction approaches to also observe the final frame.

More broadly, once we consider conditioning on the goal frame, we can devise new types of GCP models that more efficiently leverage the hierarchical structure present in real-world event sequences (Fig. 1, right). Just as coarse-to-fine image synthesis (Karras et al., 2017) generates a high-resolution image by iteratively adding details to a low-resolution image, we can synthesize a temporally downsampled video in the form of sequences of keyframes, and fill it in iteratively. We propose to

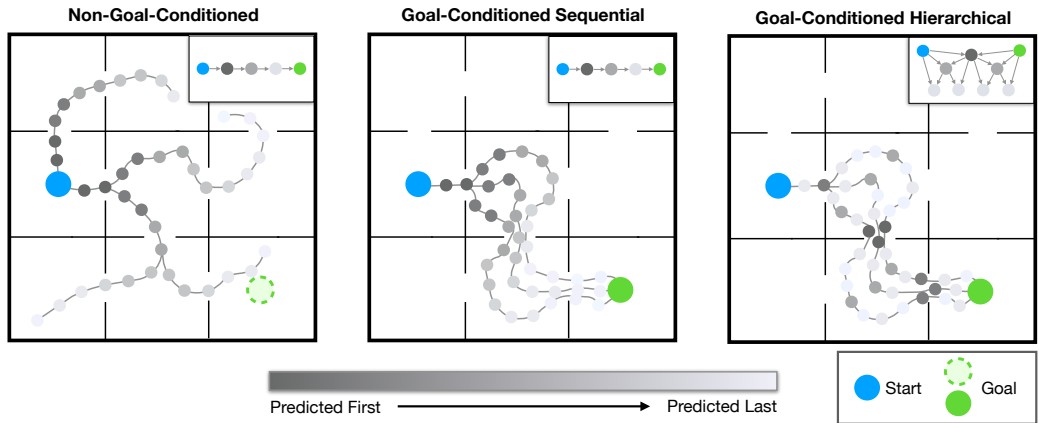

Figure 1: We study the task of prediction towards a given goal. When conditioning predictions on the goal, the distribution of possible trajectories reduces significantly (**left** vs. **middle**), making the modeling task easier and therefore allowing to scale prediction to much longer time horizons. Additionally, such goal-conditioned prediction enables more efficient, hierarchical prediction schemes (**right**).

implement this as a tree-structured GCP: the start-goal input frames serve as the initial downsampled sequence and new frames are recursively added until the full video is generated. Intuitively, this exploits the hierarchical temporal structure of natural video by breaking a long sequential procedure into its constituent steps, as illustrated in Fig 1. However, procedural steps do not all occur on a regularly spaced schedule or last for equal lengths of time. To model this, we further propose to allow the model to select which frames to generate at each level in the tree.

In our experiments, all GCP variants successfully generate longer and higher-quality video than has been demonstrated with standard auto-regressive video prediction models, which only utilize the starting frames for context. Furthermore, we show that tree-structured GCPs are more parallelizable than auto-regressive models, leading to very fast training and inference. We show that we can train tree-structured GCPs on videos consisting of thousands of frames. We also study the applications of GCPs, demonstrating that they can be utilized to enable prediction-based control in simulated imitation learning scenarios. In these settings, the GCP models can be trained without access to demonstrator actions, and can synthesize visual plans directly from start and goal images, which can then be tracked using an inverse model.

## 2 RELATED WORK

**Video generation.** Several existing neural video generation approaches either generate the entire video from scratch (Vondrick et al., 2016; Clark et al., 2019) or conditioned on the beginning of the video (Ranzato et al., 2014; Srivastava et al., 2015; Mathieu et al., 2015; Finn et al., 2016; Oh et al., 2015). Some approaches also model the uncertainties inherent in video prediction through variational inference (Denton & Fergus, 2018; Villegas et al., 2019; Xue et al., 2016; Lee et al., 2018; Larsen et al., 2016; Babaeizadeh et al., 2018). As argued above, video prediction approaches struggle to synthesize videos longer than a few seconds due to the ill-defined nature of this task. To rectify this, we propose a different class of models that conditions the prediction on both the first and the last frame of the desired video. This is related to prior work on neural video interpolation. However, such work has focused on short-term interpolation, often using models based on optical flow. Liu et al. (2017); Jiang et al. (2018) propose techniques that transform the start and the goal images via flow fields predicted by a convolutional neural network (CNN). Niklaus et al. (2017a;b) transform the start and the goal images with convolutional kernels predicted with a CNN. These methods rely on simple warping-based techniques that, while well-suited for short-term video in-filling, are not effective for generating longer video sequences. In contrast, we propose a powerful goal-conditioned video prediction technique that scales to videos as long as a minute in length. We compare against video interpolation in our experiments.

**Visual imitation and planning.** We show an application of GCP for imitation learning in the case where we have expert data *without* actions. Prior methods for solving the visual imitation learning problem have tried to learn reward functions from demonstrations to train reinforcement learning

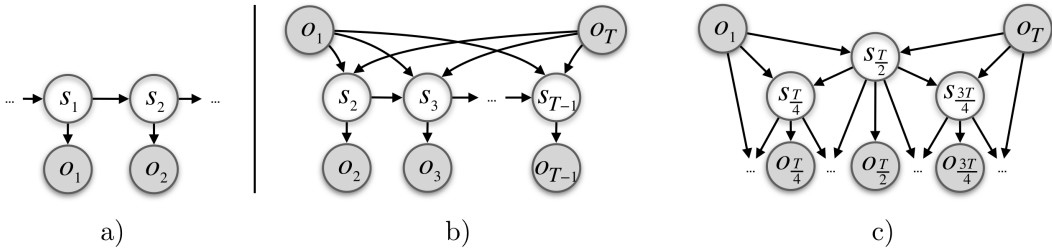

Figure 2: Graphical models for state-space video generation. **Left:** video continuation. **Right:** the proposed goal-conditioned predictors (GCPs). b) a sequential goal-conditioned prediction. c) a hierarchical goal-conditioned predictor. Shaded circles denote observations, and others are unobserved latent states.

agents to perform imitation (Sermanet et al., 2016; Peng et al., 2018; Sermanet et al., 2018), suffering from high sample complexity, and exploration problems of reinforcement learning. Our proposed goal-conditioned imitation framework, does not need to perform reinforcement learning and also leverages expert demonstrations to overcome the exploration problem. Visual goal-directed behavior can also be achieved using visual planing methods, such as visual foresight (Ebert et al., 2018) or Causal InfoGan (Kurutach et al., 2018). Unlike GCP-based imitation however, these methods do not attempt to imitate the expert, but instead compute plans internally by sampling imaginary states or actions, exhibiting difficulties for longer control horizons as targeted with our proposed method. Inverse models such as zero-shot visual imitation proposed by Pathak et al. (2018) require access to actions and are therefore insufficient to work in the action-free imitation setting.

## 3 GOAL-CONDITIONED PREDICTION

In this section, we formalize the goal-condition prediction problem, and propose several models for goal-conditioned prediction, including both auto-regressive models and tree structured models. To define the goal-conditioned prediction problem, consider a sequence of observations $[o_1, o_2, ...o_T]$ of length $T$. Standard forward prediction approaches observe the first $k$ observations and synthesize the rest of the sequence, i.e., they model $p(o_{k+1}, o_{k+2}, \ldots o_{T-1}|o_1, o_2, \ldots o_k)$. Instead, we would like our goal-conditioned predictors to produce intermediate observations given the first and last elements in the sequence. In other words, they must model $p(o_2, o_3, \ldots o_{T-1}|o_1, o_T)$. We propose several designs of goal-conditioned predictors that operate in learned compact state spaces for scalability and accuracy. Fig 2 shows schematics of these GCP designs, as well as of a standard forward predictor.

### 3.1 GOAL-CONDITIONED SEQUENTIAL PREDICTION

We first present a simple auto-regressive model for goal-conditioned prediction. In this model, we predict latent state representations sequentially in chronological order, from the start to the end, with the prediction at each point in time conditioned on the first and final observations as well as the previous latent state. The resulting model (GCP-sequential, shown in Fig 2, b) can be factorized as follows:

$$p(o_2, o_3, \ldots o_{T-1}|o_1, o_T) = \int p(o_2|s_2)p(s_2|o_1, o_T) \prod_{t=3}^{T-1} p(o_t|s_t)p(s_t|s_{t-1}, o_1, o_T)ds_{2:T-1} \quad (1)$$

We show in Sec 3.4 that this model is simple to implement, and can build directly on previously proposed auto-regressive video prediction models. However, its computational complexity scales with the sequence length, as every state must be produced in sequence. Furthermore, this approach does not account for the hierarchical structure of natural video, which contains events and sub-events, as depicted in Fig 1.

### 3.2 Goal-Conditioned Prediction by Recursive Infilling

To account for this hierarchical structure and also devise a computationally more efficient method, we now design a tree-structured GCP model.

Suppose that we have an intermediate state prediction operator $p(s_t|\text{pa}(t))$ that produces an intermediate state $s_t$ halfway in time between its two parent states $\text{pa}(t)$. Then, consider the following alternative generative process for goal-conditioned prediction: at the beginning, the observed first and last frames are encoded into the latent state space as $s_1$ and $s_T$, and the prediction operator $p(s_t|\text{pa}(t))$ generates $s_{0.5T}$. The same operator may now be applied to two new sets of parents $(s_1, s_{0.5T})$ and $(s_{0.5T}, s_T)$. As this process continues recursively, the intermediate prediction operator fills in more and more temporal detail until the full video is synthesized.

Fig 2, c depicts this model, which we call GCP-tree, since it has a tree-like shape where each predicted state is dependent on its right and left parents, starting with the start and the goal. GCP-tree factorizes the goal-conditioned video generation problem as:

$$p(o_2, o_3, \ldots o_{T-1}|o_1, o_T) = \int p(s_1|o_1)p(s_T|o_T) \prod_{t=2}^{T-1} p(o_t|s_t)p(s_t|\text{pa}(t))ds_{2:T-1}. \qquad (2)$$

**Computational efficiency.** While the sequential forward predictor performs $\mathcal{O}(T)$ sequential operations to produce a sequence of length T, the hierarchical prediction can be more efficient due to parallelization. As the depth of the tree is $\lceil \log T \rceil$, it only requires $\mathcal{O}(\log T)$ sequential operations to produce a sequence, assuming all operations that can be conducted in parallel are parallelized perfectly. We therefore batch the branches of the tree and process them in parallel at every level to utilize the benefit of efficient computation on modern GPUs. We note that the benefits of the GCP-tree runtime lie in parallelization, and thus diminish with large batch sizes, where the parallel processing capacity of the GPU is already fully utilized. We notice that, when predicting sequences of 500 frames, GCP-sequential can use up to 4 times bigger batches than GCP-Tree without significant increase in runtime cost.

**Adaptive binding.** We have thus far described the intermediate prediction operator as always generating the state that occurs halfway in time between its two parents. While this is a simple and effective scheme, it may not correspond to the natural hierarchical structure in the video. For example, in the navigation example in Figure 1, we might prefer the first split to correspond to traversing the bridge, which partitions the prediction problem into two largely independent halves. We can design a version of GCP-tree that allows the intermediate frame predictor to select which of the several states between the parents to predict, each time it is applied. In other words, the predicted state might *bind* to one of many observations in the sequence. In this more versatile model, we represent the time steps of the tree nodes with discrete latent variable $w$ that selects which nodes bind to which frames: $p(o_t|s_{1:N}, w_t) = p(o_t|s_{w_t})$. We can then express the prediction problem as:

$$p(o_{2:T-1}|o_1, o_T) = \int p(s_1|o_1)p(s_N|o_T) \prod_n p(s_n|\text{pa}(n)) \prod_{t=2}^{T-1} p(o_t|s_{1:N}, w_t)p(w_t)ds_{1:N}dw_{2:T-1}$$

$$(3)$$

Appendix C shows an efficient inference procedure for $w$ based on dynamic programming.

### 3.3 Latent variable models for GCP

To build powerful probabilistic models for goal-conditioned prediction, we propose to model the stochasticity with a per-frame latent variable $z$ (see Fig 3), inspired by prior work on forward prediction Buesing et al. (2018); Denton & Fergus (2018). To train the models, we maximize a lower bound on the likelihood of the sequence that is computed using amortized variational inference. In practice, we use a weight $\beta$ on the KL-divergence term, as is common in amortized variational inference (Higgins et al., 2017; Alemi et al., 2018; Denton & Fergus, 2018).

$$\ln p(o_{2:T-1}|o_{1,T}) \geq \mathbb{E}_{q(z_{2:T-1}|x)} \left[ \ln p(o_{2:T-1}|o_{1,T}, z_{2:T-1}) \right] - \beta KL \left[ q(z|o_{1:T}) \mid\mid p(z_{2:T-1}|o_{1,T}) \right].$$

$$(4)$$

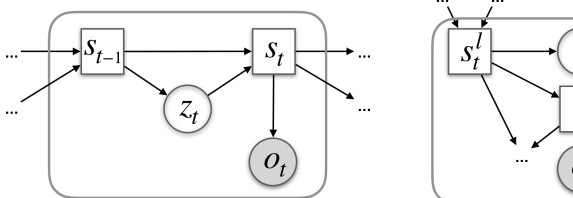

Figure 3: Proposed latent variable models. Circles represent stochastic variables, shaded circles represent variables observed at training and squares represent deterministic variables. **Left:** Sequential prediction. **Right:** Hierarchical prediction.

### 3.4 ARCHITECTURES FOR GOAL-CONDITIONED PREDICTION

Having introduced the probabilistic framework of goal-conditioned prediction, and several instantiations of that framework as graphical models, we now describe how these graphical models can be implemented as deep neural networks, to make it possible to predict sequences of high-dimensional observations $o_{1:T}$, such as images. Our predictive models depicted in Fig. 3 consists of the following parts: the prior $p(z_t|s_t)$, the deterministic recurrent predictor $p(s_t|z_t, \text{pa}(t))$, and the decoding distribution $p(o_t|s_t)$. Additionally, we use an amortized variational inference network $q(z_t|o_{1:T})$.

We parameterize these distributions as follows. The prior is a diagonal covariance Gaussian whose parameters are predicted with a multi-layer perceptron (MLP). The recurrent predictor $p(s_t|z_t, \text{pa}(t))$ is a long short-term memory network (LSTM, (Hochreiter & Schmidhuber, 1997)). The decoding distribution $p(o_t|s_t)$ is a unit variance Gaussian with the mean $\bar{o}_t$ predicted by a convolutional decoder. We condition the recurrent predictor on the start and goal images via first encoding them into embeddings $e_t = \text{enc}(o_t)$. We denote the corresponding embedding space of the decoder $\text{dec}(e_t)$.

The posterior distributions for each node, $q(z|o_{1:T})$, is computed using an attention mechanism over the embeddings of the evidence sequence (Bahdanau et al., 2015; Luong et al., 2015): $q(z_t) = \text{Att}(\text{enc}(o_{1:T}), \text{pa}(s_t))$. For simplicity, we reuse the same frame embeddings $e_t$ for the attention mechanism.

We found that using TreeLSTM (Tai et al., 2015) as the backbone of the hierarchical predictor significantly improved performance, perhaps due to better capturing long-term temporal dependencies. To increase the visual quality of the generated results, we also use skip-connections from the encoder of first and the last frame (Villegas et al., 2017; Denton & Fergus, 2018) to all decoded frames and a foreground-background generation procedure similar to (Wang et al., 2018). We activate the generated images with a hyperbolic tangent.

## 4 GOAL-CONDITIONED IMITATION LEARNING WITH GCP

We demonstrate a natural application of GCP for imitation learning, where the goal is to learn a *controller* that can select actions that reach a user-indicated goal observation. We can utilize a GCP trained on expert optimal behavior to generate a sequence of predicted observations that must occur in optimal trajectories between the current observation and the goal, i.e., a *plan*. Crucially, such a GCP can be trained on data that is not annotated with any actions – for example, a robot could learn behaviors from raw videos downloaded from YouTube. After the predictions are generated, we still need to actually select the action to take, but this is now a much simpler problem, since the GCP predicts the very next observation, and can be accomplished using a one-step *inverse model*.

Formally, at each time step, our imitation learning agent receives the current observation $o_t$ and a goal observation $o_T$. The GCP model outputs a sequences of observations $\hat{o}_t...\hat{o}_T$ leading from the start to the goal. To infer the actions necessary to execute this plan, we train a separate inverse model that estimates $p(a|o, o')$ – the probability that the action $a$ will lead from observation $o$ to $o'$. This model can be trained on *any* dataset that includes actions, including random behavior, since it does not need to perform any long-horizon reasoning. In our implementation, an inverse model is trained in a self-supervised fashion on trajectories from a random controller, without any expert supervision, while the GCP model is trained on expert data but without access to the expert's actions.

Human 3.6

Pick&Place

3x3 Maze

10x10 Maze

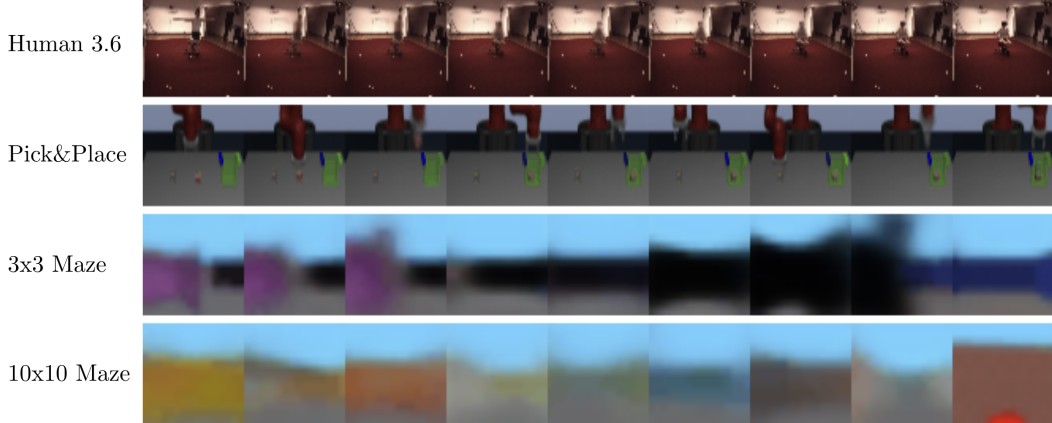

Figure 4: Prior samples from GCP-tree on the four datasets: Human 3.6, pick&place, 3x3 Maze and 10x10 Maze. Each sequence is subsampled to 9 frames.

The complete imitation method is summarized in Algorithm 1. Instead of targeting the predicted observations $\hat{o}_t$ with an inverse model, it is also possible to train the inverse model to infer actions based on the encodings of the observations $\hat{o}_t = dec(\hat{e}_t)$. The advantage of using the encodings is that uncertainties and inaccuracies induced in the image generation process cannot negatively affect the inverse model's performance.

---
**Algorithm 1** Goal Conditioned Imitation Learning
---
1: **Inputs:** Goal-conditioned predictor $g$, inverse model $p(a|s, s')$, goal observation $o_T$
2: **while** True **do**
3:     $\hat{o}_t...\hat{o}_T = g(o_t, o_T)$
4:     **for** $i = 0...n_{replan} - 1$ **do**
5:         Compute $a_t = \arg\max_a p(a|\hat{o}_i, o_T)$
6:         Execute action $a_t$.
---

## 5 EXPERIMENTAL EVALUATION

The aim of our experiments is study the following questions: 1) Are goal-conditioned models able to generate long-horizon video sequences? 2) Does tree-structured prediction improve efficiency for long-horizon video prediction? 3) Does goal-conditioned prediction enable long-term imitation without access to the expert's actions?

**Datasets.** We evaluate our model on the three datasets depicted in Fig. 5). First, the Human 3.6M dataset (Ionescu et al., 2013) is a visually complex dataset of human actors that is commonly used for evaluation of visual predictive models. We further design two simulated long-term video datasets. The *pick&place* dataset contains videos of a simulated Sawyer robot arm placing objects into a bin. Demonstrations are collected using a rule-based policy that has access to the underlying world state. Our third dataset, *Maze*, is collected in an environment based on the Gym-Miniworld (Chevalier-

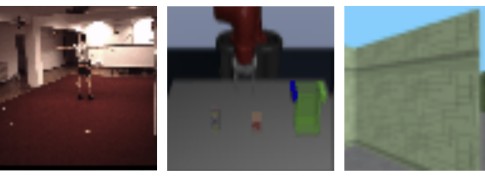

Figure 5: Datasets used for our evaluation of goal-conditioned prediction. **Left to right**: Human 3.6M (64×64px), pick&place dataset (64×64px) and Maze dataset (16×16px)

Boisvert, 2018) simulator which consists of a number of sparsely connected rooms located on a grid. The maze layout is constructed such that one single path exists between every pair of rooms and the

Table 1: Prediction performance. Pick&Place data are 80 frames 64x64, H3.6M data are 500 frames 64x64, 3x3 Maze data are 80 frames 32x32, 10x10 Maze data are 1000 frames 16x16.

| DATASET | PICK&PLACE | | HUMAN 3.6M | | 3X3 MAZE | | 10X10 MAZE | |
|---|---|---|---|---|---|---|---|---|
| METHOD | PSNR | SSIM | PSNR | SSIM | PSNR | SSIM | PSNR | SSIM |
| GCP-SEQUENTIAL | **34.45** | **0.965** | 27.57 | 0.924 | **20.83** | **0.664** | 17.08 | 0.48 |
| GCP-TREE | **34.34** | **0.965** | **28.34** | **0.928** | **20.80** | 0.652 | **17.29** | **0.495** |
| DVF | 26.15 | 0.858 | 26.74 | 0.922 | 15.798 | 0.485 | 14.50 | 0.428 |
| CIGAN | 21.16 | 0.613 | 16.89 | 0.453 | 15.57 | 0.415 | 13.81 | 0.331 |
| SVG | 31.41 | 0.960 | - | - | 19.2 | 0.600 | 17.01 | **0.496** |
| WICHERS ET AL. (2018) | 19.37 | 0.815 | - | - | 12.38 | 0.50 | - | - |

wall texture of each room is unique. Demonstrations between random start and goal positions are collected using the probabilistic roadmap (PRM) planner (Kavraki et al., 1996), leading to demonstrations with substantial noise. Due to its flexible layout the maze dataset can scale to arbitrarily long-horizon tasks. We evaluate our method on a 3×3 room and a 10×10 room layout.

We use 64×64px spatial resolution for Human 3.6M and pick&place and train on sequences of 500 frames and 80 frames respectively. For Maze, we generate sequences up to 100 frames on the 3×3 layout and 1000 frames for the 10×10 rooms. Due to the very long sequences we evaluate on a resolution of 16×16px.

## 5.1 GOAL-CONDITIONED PREDICTION

We compare GCP-sequential and GCP-tree to a state-of-the-art deep video interpolation method, DVF (Liu et al., 2017)[1] and a method for generation of visual plans, CIGAN Kurutach et al. (2018) (see Tab. 1). Following the standard procedure for evaluation of stochastic prediction models, we report top-of-100 Peak Signal-to-Noise Ratio (PSNR) and Structural Similarity Metric (SSIM). We observe that the proposed goal-conditioned prediction

Table 2: Ablation of prediction performance on pick&place

| METHOD | PSNR | SSIM |
|---|---|---|
| TREE | 34.34 | 0.965 |
| TREE W/O SKIPS | 32.64 | 0.955 |
| TREE W/O LSTM | 31.44 | 0.947 |

models outperform the video interpolation baseline by a large margin. We find that the interpolation method fails to learn meaningful long-term dynamics and instead merely blends over between the start and the goal image (see Fig. 6). CIGAN, which uses latent space interpolation, similarly struggles to capture long-term transformations, predicting unphysical changes in the scene. In contrast, both GCP-sequential and GCP-tree learn to predict rich scene dynamics in between distant start and goal frames, synthesizing sequences that traverse tens of different rooms (see Fig. 4). We attribute these results to the more powerful stochastic latent variable model our methods employ.

We also include a comparison to Stochastic Video Generation (SVG, Denton & Fergus (2018)), a prior approach that employs a stochastic latent variable model, but without goal conditioning. As expected, without the goal information the prediction problem is too underconstrained and the resulting predictions rarely match the ground truth sequene. We show that incorporating goal information leads to directed predictions of higher accuracy across all datasets. We provide evaluations of all models on the perceptual metrics *Fréchet Video Distance* (FVD, Unterthiner et al. (2018)) and *Learned Perceptual Image Patch Similarity* (LPIPS, Zhang et al. (2018)) in the appendix, Tab. 4.

To further inspect the properties of the proposed approach, we ablate different architectural choices in Tab. 2 using the tree-structured predictor. We show that both skip connections from the parents and start and goal as well as recurrence in the predictive module are essential to achieve high prediction performance.

### 5.1.1 RUNTIME COMPARISON

---

[1]We note that all methods including DVF were trained on the training split of the datasets they were tested on to allow for fair comparison.

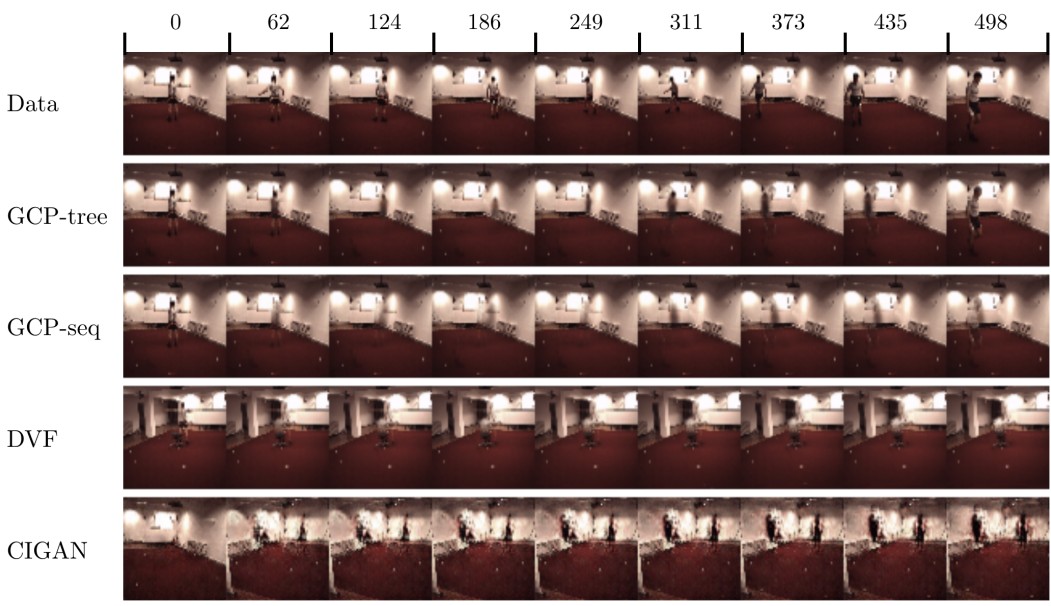

Figure 6: Predictions on Human 3.6M. We see that the GCP models are able to faithfully capture the human trajectory. The optical flow-based method (DVF) captures the background but fails to generate complex motion needed for long-term goal-conditioned prediction. Causal InfoGan also struggles to capture the structure of these long sequences and produce implausible interpolations.

One advantage of the hierarchical prediction approach of GCP-Tree over the frame-by-frame prediction of GCP-sequential is that the former allows for heavy parallelization of computation, leading to substantially reduced sequential computational complexity (see Sec. 3.2). We validate that we can realize these runtime benefits by comparing the time required for each training iteration of forward and tree-structured prediction across different sequence lengths (see Fig. 7). We find that, especially for very long sequences, GCP-Tree has a much lower iteration time, enabling more efficient training on long-horizon prediction tasks.

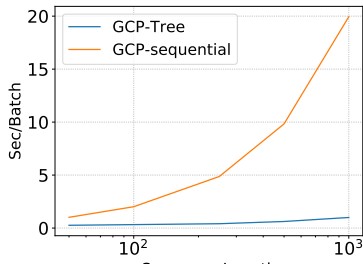

Figure 7: Runtime of GCP for different sequence lengths on 10x10 Maze ($16 \times 16$px) with a batch size of four. Experiments were performed on a standard NVIDIA P100 GPU.

### 5.2 BOTTLENECK DISCOVERY

We evaluate the ability of the model with adaptive binding to learn the structure of the dataset. To do so, we set the decoder distribution variance of the first two layers of the tree to 50, forcing this node to bind to the frame for which the prediction is the most confident. In Fig. 8, we see that this indeed causes the model to bind the top nodes to the frames that is the easiest to predict – e.g. the semantic bottleneck where the robot arm is just about to drop the object in the bin. We found that the two other nodes find similar bottleneck points. Further details of this experiment are described in the Appendix C. Experimentally, we found that adaptive binding did not substantially improve quantitative accuracy of the predictions on our datasets, though the ability to discover meaningful bottlenecks may itself be a useful property in future work, as discussed in the literature on hierarchical RL (Barto & Mahadevan, 2003) or exploration (Goyal et al., 2019).

### 5.3 GOAL-CONDITIONED CONTROL

We evaluate whether goal-conditioned visual prediction is suitable for imitating long-term, goal-directed expert behavior without access to the expert's actions. We first collect a dataset of 30k expert

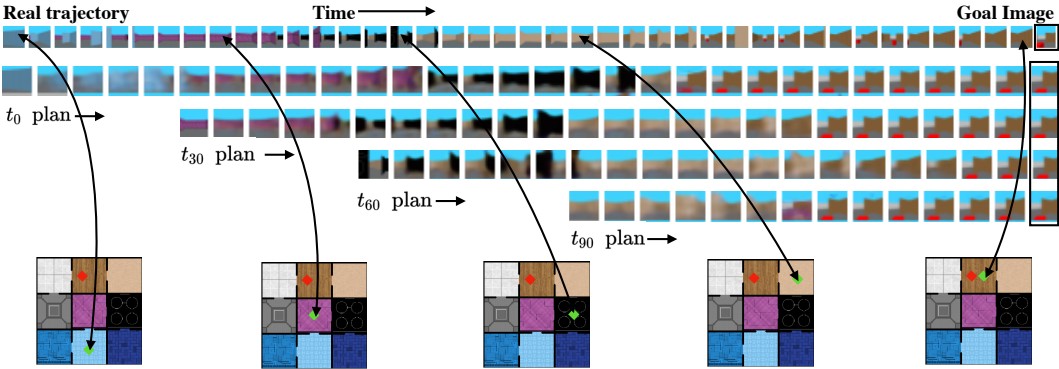

Figure 9: Illustration of the agent successfully following plans generated by GCP-sequential model through 5 rooms in succession. The agent's position in the map is marked with the red diamond, the goal is marked in green; time progresses from left to right; re-planning is performed every 5 steps. We show temporally subsampled plans and executed sequence; the full trajectory has a length of 200 steps.

trajectories with up to 100 frames in the $3\times3$ Maze environment. For planning and execution we follow the procedure detailed in Sec. 4. We train the inverse model that is used to follow the visual plans using 20k trajectories of 15 frames each, collected by taking random actions from random initial positions in the maze.

We quantitatively evaluate the performance of goal-conditioned prediction for visual imitation learning by randomly sampling 100 start and goal positions in the map and measuring the difference between the initial and final distance to the goal after the episode finished, computed as the shortest path distance through the maze. We compare GCP-sequential to two alternative visual planning methods: visual foresight (Ebert et al., 2018) and Causal InfoGan (CIGAN, Kurutach et al. (2018)). We additionally report performance of a behavior cloning approach, but note that it requires ground truth action annotations for the demonstration dataset while all other approaches can learn from pure visual demonstrations only.

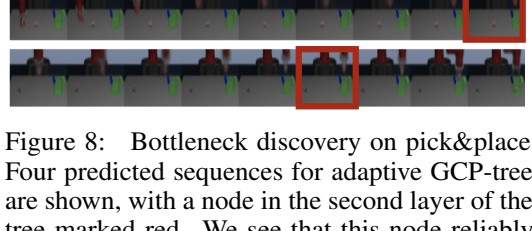

Figure 8: Bottleneck discovery on pick&place. Four predicted sequences for adaptive GCP-tree are shown, with a node in the second layer of the tree marked red. We see that this node reliably identifies and predicts a semantic bottleneck in the trajectory, which is the position where the agent is about to drop the object in the bin.

We find that none of the alternative methods for visual planning is able to scale to long-horizon tasks that require the traversal of multiple rooms. In particular, we observe that CIGAN is unable to produce coherent, long-horizon plans and instead generates smooth interpolations between start and goal image. Consequently, both prior methods fail to complete most of the tasks while GCP-sequential can generate long-horizon, coherent plans (see Fig. 9) and complete substantially more of the test tasks (see Tab. 3). Finally, behavior cloning achieves higher performance but requires access to ground truth demonstration actions at training time which can be very costly or impossible to obtain in realistic settings.

# 6 DISCUSSION AND FUTURE WORK

We presented goal-conditioned predictors (GCPs) – predictive models that generate video sequences between a given start and goal frame. GCPs must learn to understand the mechanics of the environment that they are trained in, in order to accurately predict the intermediate events that must take

place in order to bring about the goal images from the start images. GCP models not only allow for substantially more accurate video prediction than conventional models that are conditioned only on the beginning context, but also allow for novel model architectures. Specifically, we explore how, in addition to more conventional auto-regressive GCPs, we can devise tree-structured GCP models that predict video sequences hierarchically, starting with the coarsest level subgoals and recursively subdividing until a full sequence is produced.

Our experimental results show that GCPs can make more accurate predictions. We also demonstrate that they can be utilized in an imitation learning scenario, where they can learn behaviors from video demonstrations without example actions. Imitation from observations, without actions, is applicable in a wide range of realistic scenarios. For example, a robot could learn the mechanics of cooking from watching videos on YouTube (Damen et al., 2018), and then use this model to learn how to cook on its own. We hope that the imitation framework presented in our work can be a step in towards effectively leveraging such data for robotic control.

Table 3: Control Performance on the Maze task

| METHOD | SUCCESS |
|---|---|
| GCP-SEQ | **59%** |
| CIGAN | 10% |
| vMPC | 7% |
| BC | 87% |

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

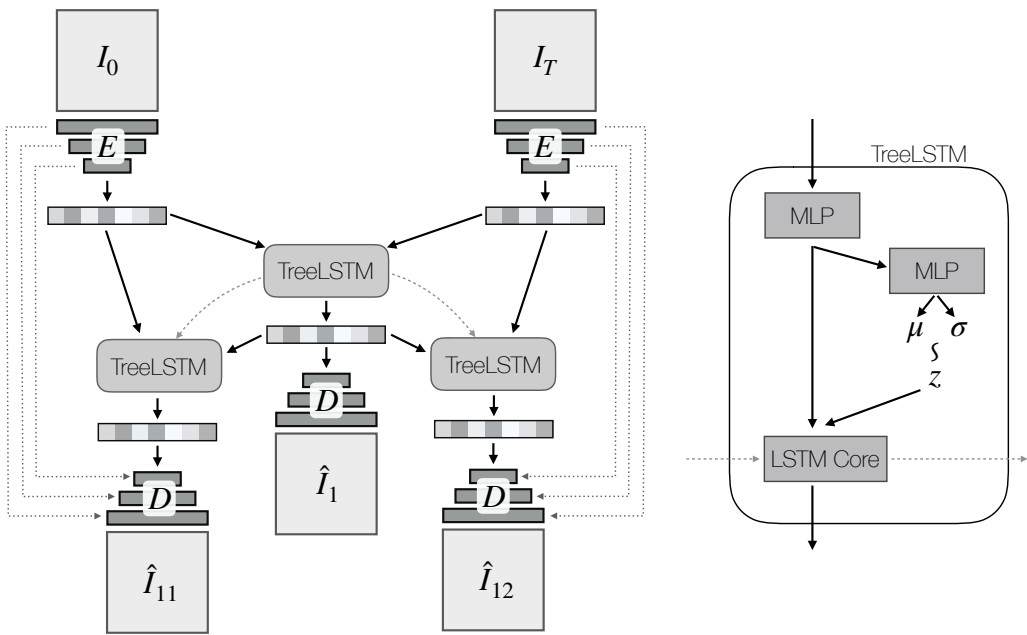

Figure 10: Schematic visualization of the model architecture for GCP-Tree (first two layers depicted). Dashed lines represent passing of the LSTM state from one iteration to the next, dotted lines depict skip connections (skip connections to the decoder in the first iteration omitted for clarity).

## A    DATA PROCESSING

For the Human 3.6 dataset, we downsample the original videos to 64 by 64 resolution. We obtain videos of length of roughly 800 to 1600 frames, which we randomly crop in time to 500-frame sequences.

We split the Human 3.6 into training, validation and test set by correspondingly 95%, 5% and 5% of the data. On the TAP dataset, we use 48949 videos for training, 200 for validation and 200 for testing.

## B    ARCHITECTURE

We use a convolutional encoder and decoder similar to the standard DCGAN discriminator and generator architecture respectively. The latent variables $z_n$ as well as $e_n$ are 32-dimensional. All hidden layers in the Multi-Layer Perceptron have 32 neurons. We add skip-connections from the encoder activations from the first image to the decoder for all images. For the inference network implemented with attention, we found it beneficial to use a 2-layer 1D temporal convolutional network that adds temporal context into the latent vectors $e_t$ before attention. For the recursive predictor that predicts $e_n$, we found it crucial for the stability of the training to activate $e_n$ with hyperbolic tangent (tanh), and use group normalization Wu & He (2018). We observed that without this, the magnitude of activations can explode in the lower levels of the tree and conjecture that this is due to recursive application of the same network. We found that batch normalization Ioffe & Szegedy (2015) does not work as well as group normalization for the recursive predictor and conjecture that this is due to the activation distributions being non-i.i.d. for different levels of the tree. We use batch normalization in the convolutional encoder and decoder, and use local per-image batch statistics at test time.

**Hyperparameters.**    For each method and dataset, we performed a manual sweep of the hyperparameter $\beta$ in the range from $1e{-}0$ to $1e{-}4$. The convolutional encoder and decoder both have five layers. We use the Rectified Adam optimizer (Liu et al., 2019; Kingma & Ba, 2015) with $\beta_1 = 0.9$ and $\beta_2 = 0.999$, batch size of 16 for GCP-sequential and 4 for GCP-tree, and a learning rate of

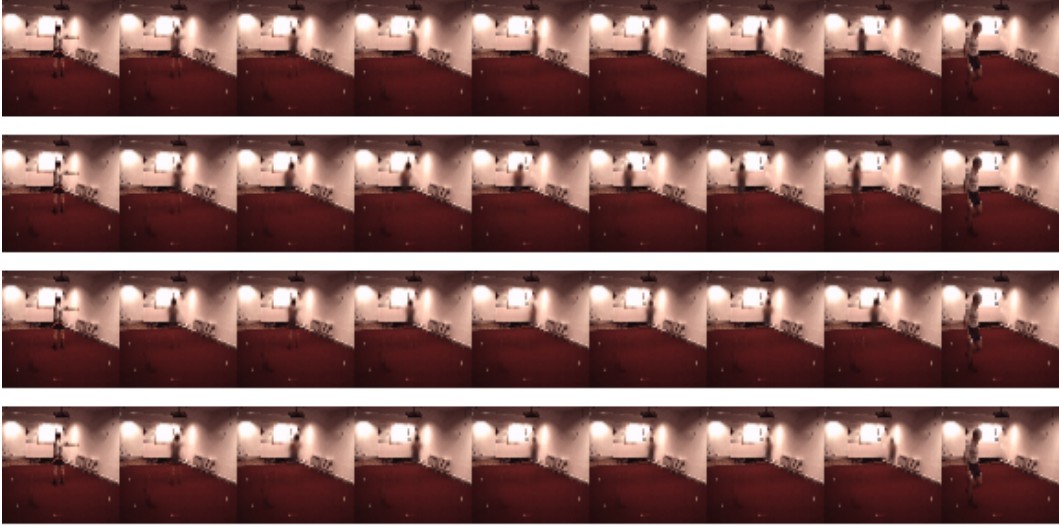

Figure 11: Prior samples from GCP-tree on the Human 3.6M dataset. Each row is a different prior sample conditioned on the same information.

$2e-4$. On each dataset, we trained each network for the same number of epochs on a single high-end NVIDIA GPU.

## C  ADAPTIVE BINDING WITH DYNAMIC PROGRAMMING

To optimize the model with adaptive binding, we perform variational inference on both $w$ and $z$:

$$\log p(x) \geq \mathbb{E}_{q(z,w)}[p(x|w,z)] - D_{KL}(q(z|x)||p(z)) - D_{KL}(q(w|x,z)||p(w)). \qquad (5)$$

To infer $q(w|x,z)$, we want to produce a distribution over possible alignments between the tree and the evidence sequence. Moreover, certain alignments, such as the ones that violate the ordering of the sequence are forbidden. We define such distribution over aligment matrices $A$ via Dynamic Time Warping. We define the energy of an alignment matrix as the cost, and the following distribution over alignment matrices:

$$p(A|x,y) = \frac{1}{Z}e^{-A*c(x,z)},$$

where $Z = \mathbb{E}_A[e^{-A*c(x,z)}]$, and $c$ is the MSE error between the ground truth frame $x_t$ and the decoded frame associated with $z_n$. We are interested in computing marginal edge distributions $w = \mathbb{E}_A[A]$. Given these, we can compute the reconstruction error efficiently. We next show how to efficiently compute the marginal edge distributions.

Given two sequences $x_{0:T}, z_{0:N}$, denote the partition function of aligning two subsequences $x_{0:i}, z_{0:j}$ as $f_{i,j} = \sum_{A \in \mathcal{A}_{0:i,0:j}} e^{-A*c(x_{0:i},z_{0:j})}$. Cuturi & Blondel (2017) shows that these can be computed efficiently as:

$$f_{i,j} = c(x_i, z_j) * (f_{i-1,j-1} + f_{i-1,j}).$$

Futhermore, denote the partition function of aligning $x_{i:T}, z_{j:N}$ as $b_{i,j} = \sum_{A \in \mathcal{A}_{i:T,j:N}} e^{-A*c(x_{i:T},z_{j:N})}$. Analogously, we can compute it as:

$$b_{i,j} = c(x_i, z_j) * (b_{i+1,j+1} + b_{i+1,j}).$$

**Proposition 1** *The total unnormalized density of all alignment matrices including the edge $(i,j)$ can be computed as $e_{i,j} = f_{i,j} * b_{i,j}/c(x_i, z_j) = c(x_i, z_j) * (f_{i-1,j-1} + f_{i-1,j}) * (b_{i+1,j+1} + b_{i+1,j})$. Moreover, the probability of the edge $(i,j)$ can be computed as $w_{i,j} = e_{i,j}/Z$.*

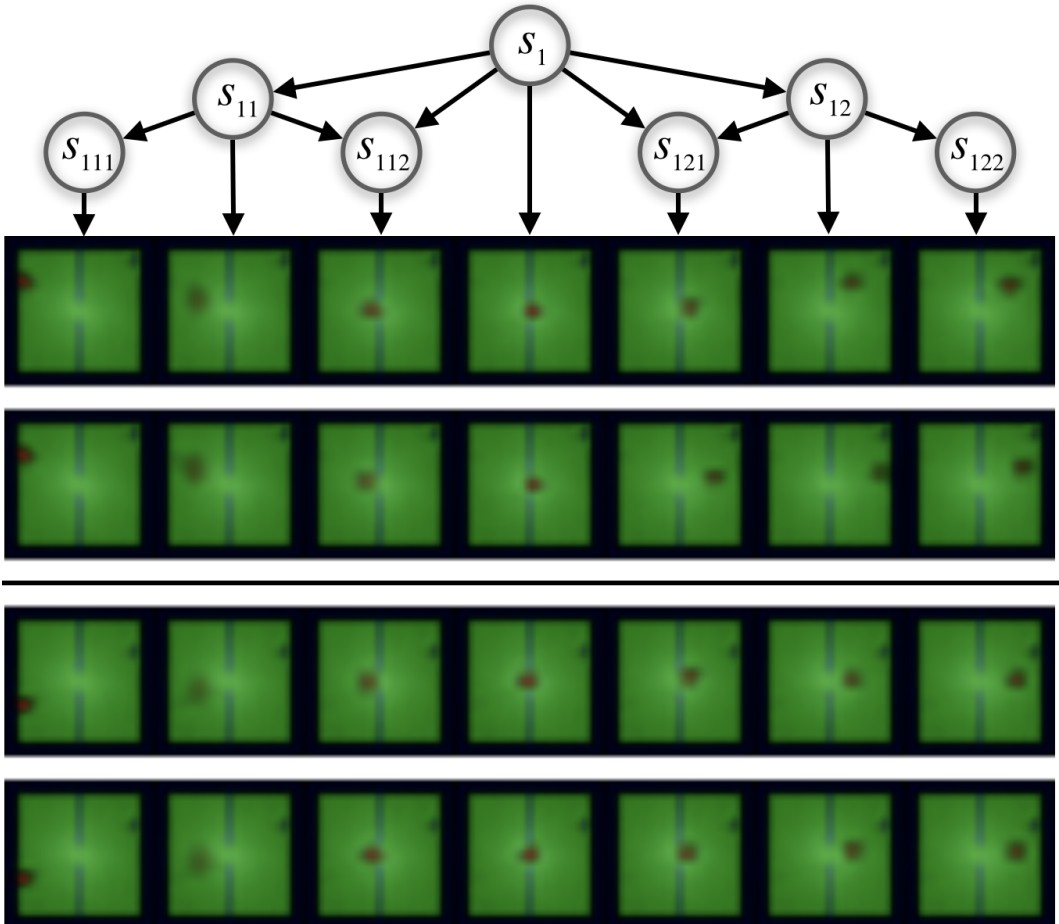

Figure 12: Prior samples from GCP-tree on the 2D maze dataset. Each pair of rows shows two samples given the same context.

Table 4: Prediction performance on perceptual metrics. Pick&Place data are 80 frames 64x64, H3.6M data are 500 frames 64x64, 3x3 Maze data are 80 frames 32x32, 10x10 Maze data are 1000 frames 16x16.

| DATASET | PICK&PLACE | | HUMAN 3.6M | | 3X3 MAZE | | 10X10 MAZE | |
|---|---|---|---|---|---|---|---|---|
| METHOD | FVD | LPIPS | FVD | LPIPS | FVD | LPIPS | FVD | LPIPS |
| GCP-SEQUENTIAL | **328.9** | **0.02** | 1541.8 | **0.06** | **359.2** | **0.17** | **472.7** | **0.26** |
| GCP-TREE | 430.3 | **0.02** | **1314.3** | **0.05** | 535.0 | **0.18** | **473.7** | **0.26** |
| DVF | 2879.9 | 0.06 | 1704.6 | **0.05** | 470.7 | 0.28 | 529.5 | 0.32 |
| CIGAN | 3252.6 | 0.12 | 2528.5 | 0.17 | 756.0 | 0.36 | 486.9 | 0.33 |
| SVG | 713.6 | **0.01** | - | - | 498.1 | 0.19 | 488.5 | 0.28 |

Proposition 1 enables us to compute the expected reconstruction cost in quadratic time:

$$w * c(x, y).$$

