# OpenReview forum: "Goal-Conditioned Video Prediction"
_ICLR.cc/2020/Conference — Reject_

### Official Review · AnonReviewer3 · 2019-10-23
**Official Blind Review #3**

**Rating:** 3

**Review:**

This paper reformulates video prediction problem by conditioning the prediction on the start and end (goal) frame. This essentially changes the problem from extrapolation to interpolation which results in higher quality predictions.

The motivation behind the paper is not clear. First of all, the previous work in video predicted is typically formulated as "conditioned frame prediction" where the prediction of the next frame is conditioned on "a set of context frames" and there is no reason why this set cannot contain the goal frame. Their implementation, however, is motivated by their application and therefore these models are usually only conditioned on the start frames. Unfortunately, besides the reverse planning in imitation learning, the authors did not provide a suite of applications where such a model can be useful. Hence, I think the authors should answer these two questions to clear up the motivation:
1. Why conditioning on the goal frame is interesting? It specifically helps to provide more concrete details than getting from Oakland to San Fransico.
2. Where the current conditional models suffer by conditioning on the goal image?

More experiments are required to support the claims of the paper as well.
Given my point regarding context frames, a more fair experiment would be to compare the proposed method with them when they are conditioned on the goal frame as well. This explicitly has been avoided in 5.1.
 The used metrics are not a good evaluation metric for frame prediction as they both do not give us an objective evaluation in the sense of the semantic quality of predicted frames. The authors should present additional quantitative evaluation to show that the predicted frames contain useful semantic information. FVD and Inception score come to my mind as good candidates.

On quality of writing, the paper is well written but it can use a figure that demonstrates proposed architecture. The authors provided the code which is always a plus.

In conclusion, I believe the impact of the paper, in the current form, is marginal at best and for sure does not meet the requirements for a prestigious conference such as ICLR. However, a more clear motivation, a  concrete set of goals and claims, as well as more comprehensive experiments,  can push the quality above the bar.

**Experience Assessment:**

I have published in this field for several years.

**Review Assessment: Checking Correctness Of Derivations And Theory:**

N/A

**Review Assessment: Checking Correctness Of Experiments:**

I carefully checked the experiments.

**Review Assessment: Thoroughness In Paper Reading:**

N/A

---

> ### Author Response · Authors · 2019-11-15
> **Added FVD/LPIPS, clarified motivation**
>
> We thank the reviewer for the comments on the motivation and suggesting additional experiments. As suggested, we made the following changes:
> - In Tab 4, evaluated the compared models on FVD and LPIPS, perceptual visual quality metrics, showing that both goal-conditioned prediction models outperform all baselines across all datasets.
> - Improved the presentation of our motivation and the introductory figure.
>
> We answer the questions in detail below. Please let us know if this addresses your concern, or if you would like us to discuss this further or add additional evaluations!
>
> == 1. Why conditioning on the goal frame is interesting? ==
> A: We thank the reviewer for bringing up the important point of motivation. We revised the introductory figure to more clearly reflect our motivation, and we next provide detailed application examples for goal-conditioned prediction (GCP) that expand on the motivation in our introduction. We will integrate these arguments in the final version of the paper.
> - When controlling an agent, the goal state is often known in practice, and utilizing it for prediction should allow to construct better plans. Building such goal-conditioned agents with model-free techniques is an active area of research [1, 2, 3, 4, 5, 6, 7]. We are hopeful that building better goal-conditioned predictors will enable use of data efficient model-based techniques for such problems.
> - More generally, in many natural settings the goal of a certain process is known and we want to leverage it for video generation. An example application of GCP is a tool that allows to edit or create a video. To modify a video, a human graphics designer might simply want to change a few seconds of video, and GCP can generate the interpolations to smoothly embed the frames in the video. This problem is distinct from open-ended forward prediction as the video is constrained by the desired final frame.
> - Finally, we argue in the introduction that unconstrained prediction without a goal is often very challenging, as uncertainty increases dramatically for long time horizons. Conditioning on the goal reduces the uncertainty and makes long-horizon video prediction beyond lengths considered by prior work tractable, as our paper shows.
>
> == 2. Where the current conditional models suffer by conditioning on the goal image? ==
> We find that for long-horizon goal-conditioned prediction an expressive model that is able to handle the stochasticity in long sequences well is necessary. The two goal-conditioned prediction methods we compare to, DVF and CIGAN, are unable to handle the complexity of such prediction as they are designed for rather short sequences. This motivated our sequential latent variable approach. We note that certain prior work like Denton&Fergus’18, Lee’18, used sequential latent variable models for forward prediction and therefore one version of our proposed method, GCP-sequential, can be considered the goal-conditioned extension of this prior work. We clarified this in the manuscript.
>
> [1] Kaelbling, Leslie Pack. "Learning to achieve goals." IJCAI. 1993.
> [2] Schaul, Tom, et al. "Universal value function approximators." International Conference on Machine Learning. 2015.
> [3] Andrychowicz, Marcin, et al. "Hindsight experience replay." Advances in Neural Information Processing Systems. 2017.
> [4] Pong, Vitchyr, et al. "Temporal difference models: Model-free deep rl for model-based control." ICLR. 2018.
> [5] Nair, Ashvin V., et al. "Visual reinforcement learning with imagined goals." Advances in Neural Information Processing Systems. 2018.
> [6] Fu, Justin, et al. "Variational inverse control with events: A general framework for data-driven reward definition." Advances in Neural Information Processing Systems. 2018.
> [7] Warde-Farley, David, et al. "Unsupervised control through non-parametric discriminative rewards." ICLR. 2019.

---

### Official Review · AnonReviewer2 · 2019-10-23
**Official Blind Review #2**

**Rating:** 6

**Review:**

Summary: The following work proposes a model for long-range video interpolation -- specifically targetting cases where the intermediate content trajectories may be highly non-linear. This is referred to as goal-conditioned in the paper. They present an autoregressive sequential model, as well as a hierarchical model -- each based on a probabilistic framework. Finally, they demonstrate an application in imitation learning by introducing an additional model that maps pairs of observations (frames) to a distribution over actions that predicts how likely each action will map the first observation to the second. Their imitation learning method is able to successfully solve mazes, given just the start and goal observations.

Strengths:
-The extension to visual planning/imitation learning was very interesting
-Explores differences between sequential and hierarchical prediction models

Weaknesses/questions/suggestions:
-In addition to SSIM and PSNR, one might also want to consider the FVD and LPIPS, both which should correlate better with human perception.
-How does the inverse model in section $ p(a | o,o')$ account for the case in which multiple actions may eventually result in o -> o', given than o' is sufficiently far from o? Does the random controller need to be implemented in a specific way to handle this?
-I think a fairly important unstated limitation is that latent-variable based methods tend not to generalize well outside of their trained domain. In table 1, I assume DVF was taken off-the-shelf, but all other methods were trained specifically on H3.6M?


LPIPS: https://github.com/richzhang/PerceptualSimilarity
FVD: https://github.com/google-research/google-research/tree/master/frechet_video_distance


Overall, I think the results seem pretty promising -- most notably the imitation learning results. I hope that the authors can address some of my concerns stated above.


** Post Rebuttal:
The authors have adequately addressed my concerns regarding clarity and metrics. The current draft also better motivates the task of long-range interpolation vs short range interpolation. I maintain my original rating.

**Experience Assessment:**

I have published one or two papers in this area.

**Review Assessment: Checking Correctness Of Derivations And Theory:**

I assessed the sensibility of the derivations and theory.

**Review Assessment: Checking Correctness Of Experiments:**

I assessed the sensibility of the experiments.

**Review Assessment: Thoroughness In Paper Reading:**

I read the paper at least twice and used my best judgement in assessing the paper.

---

> ### Author Response · Authors · 2019-11-15
> **Added FVD/LPIPS evaluation, additional clarifications**
>
> We thank the reviewer for the helpful comments and suggestions. We made the following changes to the submission to address the reviewers remarks and answer the posed questions:
>
> == FVD+LPIPS metrics ==
> We added evaluation results with both metrics for all four datasets to Tab.4 in the appendix. We find that both proposed models for goal-conditioned prediction outperform video interpolation baselines as well as non-goal-conditioned prediction.
>
> == Stochastic inverse model ==
> Indeed it is possible that multiple different action sequences lead from a start state o to a goal state o’ and prior work addressed this problem by conditioning the inverse model on a stochastic latent variable to explicitly model the uncertainty over the action trajectory [1]. However, in our experiments we did not find to be an issue, because there are typically only 1-3 time steps between the current state and the next predicted target of the inverse model. This is because GCP is able to predict a dense plan for the inverse model to follow. We note that the proposed method is general and can be used with stochastic inverse models.
>
> == DVF Off-the-shelf ==
> We want to point out that *all methods* were trained from scratch on the respective domain they were tested on, i.e. we re-trained the DVF model that we used to report numbers on H3.6M using the H3.6M training set. This is to allow fair comparison to the GCP models that were trained on the same data. We did not use the off-the-shelf DVF network. We thank the reviewer for pointing out this possible confusion and we added a footnote to the revised manuscript clarifying that all models were trained from scratch.
>
> We again thank the reviewer for the helpful suggestions that improved the quality of the submission. Please let us know if there are any further questions!
>
> [1] Learning Latent Plans from Play, Lynch et al., 2019

---

### Official Review · AnonReviewer1 · 2019-10-24
**Official Blind Review #1**

**Rating:** 6

**Review:**


REFERENCES ARE LISTED AT THE END OF THE REVIEW


Summary:
This paper proposes a method for video prediction that, given a starting and ending image, is able to generate the frame trajectory in between. They propose two variations of their method: A sequential and a tree based methods. The tree-based method enables efficient frame sampling in a hierarchical way. In experiments, they outperform the used baselines in the task of video prediction. Additionally, they used the learned pixel dynamics model and an inverse dynamics model to plan actions for an agent to navigate from a starting frame to an ending frame.


Pros:
+ Novel latent method for goal conditioned prediction (sequential and hierarchical)
+ Really cool experiments on navigation using the predicted frames
+ Outperforms used baselines

Weaknesses / comments:
- Missing baseline:
The Human 3.6M experiments are missing the baseline from Wichers et al., 2018. I would be good to compare against them for better assessment of the predicted videos.

- Bottleneck discovery experiments (Figure 8):
The visualizations shown in Figure 8 are very interesting, however, I would like to see if the model is able to generate multiple trajectories from the same frame. It looks like the starting frames (left) are not the same.


Conclusion:
This paper proposes a novel latent variable method for goal oriented video prediction which is then used to enable an agent to go from point A to point B. I feel this paper brings nice insights useful for the model based reinforcement learning literature where the end goal can be guided by an image rather than predefined rewards. It would be good if the authors can include the suggested video prediction baseline from Wichers et al., 2018 in their quantitative comparisons.


References:
Nevan Wichers, Ruben Villegas, Dumitru Erhan, Honglak Lee. Hierarchical Long-term Video Prediction without Supervision. In ICML, 2018


**Experience Assessment:**

I have published one or two papers in this area.

**Review Assessment: Checking Correctness Of Derivations And Theory:**

N/A

**Review Assessment: Checking Correctness Of Experiments:**

I carefully checked the experiments.

**Review Assessment: Thoroughness In Paper Reading:**

I read the paper thoroughly.

---

> ### Author Response · Authors · 2019-11-15
> **Added Wichers'18 Comparison, added additional qualitative visualizations, updated bottleneck results**
>
> We thank the reviewer for the helpful comments and suggestions. To address the reviewers remarks we made the following improvements to the paper.
>
> == Wichers’18 ==
> As suggested by the reviewer, we trained Wichers’18 and report video prediction metrics in Tab. 1. We observe that this method struggles in our experimental setup, likely because deterministic prediction given only one conditioning frame is challenging, especially in stochastic environments. We have made an attempt at extending this baseline to the goal-conditioned setting. However, in our preliminary experiments we were not able to improve the performance over the original version. We also note that we were not able to run Wichers’18 on datasets longer than 100 frames due to computational requirements.
>
> == Multiple sampled sequences ==
> We added a visualization of multiple sequences sampled given the same start-goal frames to the appendix, Figure 10 for the Human 3.6 dataset and Figure 11 for the 2D maze dataset. We note that the original supplementary website contained examples of multiple sampled sequences for every dataset.
>
> == Bottleneck discovery ==
> To further investigate the bottleneck discovery phenomenon, we performed an experiment on the Pick&Place data, and we observe that the model reliably discovers bottlenecks in those data too. The generations are now shown in Fig. 8.
>
> We again thank the reviewer for the helpful suggestions that improved the quality of the submission. Please let us know if there are any further questions!

---

### Author Response · Authors · 2019-11-15
**Author Response: added Wichers'18 comparison, added FVD/LPIPS evaluation, updated bottleneck results, added clarifications**

We thank all reviewers for the helpful comments and suggestions. To address them we made the following changes to the manuscript:
(1) We added comparison to the video prediction model of Wichers’18 to Tab 1, showing that both our models, GCP-sequential and GCP-tree, outperform the added baseline on multiple datasets.
(2) We added evaluation with the perceptual metrics FVD and LPIPS to Tab 4 in addition to the reported standard video prediction metrics PSNR/SSIM. We show that both proposed goal-conditioned prediction models outperform all baselines on the added metrics across the four tested datasets.
(3) We extended the analysis of bottleneck discovery for hierarchical GCP to the Pick&Place dataset and find that the model is able to discover bottleneck states in the top nodes of the predicted hierarchy.
(4) We added clarifications to multiple sections of the manuscript addressing questions the reviewers raised. We updated Fig. 1 to better visualize the motivation of the approach. We updated Fig. 8 with the added bottleneck results and added Tab. 4 to the appendix to include the added comparisons and metrics. Further, we added an architecture figure to the appendix.

---

### Decision · Program_Chairs · 2019-12-19

**Decision:**

Reject

**Comment:**

The paper addresses a video generation setting where both initial and goal state are provided as a basis for long-term prediction. The authors propose two types of models, sequential and hierarchical, and obtain interesting insights into the performance of these two models. Reviewers raised concerns about evaluation metrics, empirical comparisons, and the relationship of the proposed model to prior work.

While many of the initial concerns have been addressed by the authors, reviewers remain concerned about two issues in particular. First, the proposed model is similar to previous approaches with sequential latent variable models, and it is unclear how such existing models would compare if applied in this setting. Second, there are remaining concerns on whether the model may learn degenerate solutions. I quote from the discussion here, as I am not sure this will be visible to authors [about Figure 12]: "now the two examples with two samples they show have the same door in the middle frame which makes me doubt the method learn[s] anything meaningful in terms of the agent walking through the door but just go to the middle of the screen every time."